# Similarities in DSG1 and KRT3 Downregulation through Retinoic Acid Treatment and PAX6 Knockdown Related Expression Profiles: Does PAX6 Affect RA Signaling in Limbal Epithelial Cells?

**DOI:** 10.3390/biom11111651

**Published:** 2021-11-08

**Authors:** Lorenz Latta, Igor Knebel, Constanze Bleil, Tanja Stachon, Priya Katiyar, Claire Zussy, Fabian Norbert Fries, Barbara Käsmann-Kellner, Berthold Seitz, Nóra Szentmáry

**Affiliations:** 1Dr. Rolf M. Schwiete Center for Limbal Stem Cell and Congenital Aniridia Research, Saarland University, 66421 Homburg, Germany; lorenz.latta@uks.eu (L.L.); igorknebel7@googlemail.com (I.K.); Constanze.Bleil@gmx.de (C.B.); Tanja.Stachon@uks.eu (T.S.); priya21katiyar@gmail.com (P.K.); claire.zussy@live.fr (C.Z.); 2Department of Ophthalmology, Saarland University Medical Center, 66421 Homburg, Germany; fries.fn@gmail.com (F.N.F.); Barbara.Kaesmann-Kellner@uks.eu (B.K.-K.); Berthold.Seitz@uks.eu (B.S.)

**Keywords:** aniridia, retinoic acid, PAX6, limbal epithelial cells, DSG1, ADH7, aniridia-associated keratopathy (AAK)

## Abstract

Congenital PAX6-aniridia is a rare panocular disease resulting from limbal stem cell deficiency. In PAX6-aniridia, the downregulation of the retinol-metabolizing enzymes ADH7 (All-trans-retinol dehydrogenase 7) and ALDH1A1/A3 (Retinal dehydrogenase 1, Aldehyde dehydrogenase family 1 member A3) have been described in limbal epithelial cells (LECs) and conjunctival epithelial cells. The aim of this study was to identify the role of retinol derivates in the differentiation of human LEC and its potential impact on aniridia-associated keratopathy development. Human LEC were isolated from healthy donor corneas and were cultured with retinol, retinoic acid, or pan-retinoic acid receptor antagonist (AGN 193109) acting on RARα, β, γ (NR1B1, NR1B2 NR1B3) or were cultured with pan-retinoid X receptor antagonist (UVI 3003) acting on RXR α, β, γ (retinoid X receptor, NR2B1, NR2B2, BR2B3). Using qPCR, differentiation marker and retinoid-/fatty acid metabolism-related mRNA expression was analysed. DSG1 (Desmoglein 1), KRT3 (Keratin 3), and SPINK7 (Serine Peptidase Inhibitor Kazal Type 7) mRNA expression was downregulated when retinoid derivates were used. AGN 193109 treatment led to the upregulation of ADH7, KRT3, and DSG1 mRNA expression and to the downregulation of KRT12 (Keratin 12) and KRT19 (Keratin 19) mRNA expression. Retinol and all-trans retinoic acid affect some transcripts of corneal LEC in a similar way to what has been observed in the LEC of PAX6-aniridia patients with the altered expression of differentiation markers. An elevated concentration of retinol derivatives in LEC or an altered response to retinoids may contribute to this pattern. These initial findings help to explain ocular surface epithelia differentiation disorders in PAX6-aniridia and should be investigated in patient cells or in cell models in the future in more detail.

## 1. Introduction

Retinoids act through nuclear receptors at the transcriptional level; however, the non-receptor-mediated functions of retinoids have not been extensively studied as of yet [1]. Besides their important role in development, retinoids also have the capability to affect apoptosis, the differentiation and proliferation of skin epidermal cells [2], as well as to affect the apoptosis, proliferation, surface wound healing, and keratinization of corneal epithelial cells [1,3]. A severe vitamin A deficiency leads to xerophthalmia (dry eye) and keratopathy [4].

All-trans retinoic acid (at-RA) acts through binding to the retinoic acid-receptors RAR α, β, γ (NR1B1, NR1B2 NR1B3), RXR α, β, γ (retinoid X receptor, NR2B1, NR2B2, NR2B3), and PPARα, β/δ, γ (peroxisome proliferator-activated receptors NR1C1, NR1C2, NR1C3).

Patients with a haploinsufficiency of the PAX6 (Paired box protein Pax-6) gene (*PAX6*^+/-^) suffer from complete or partial absence of the iris tissue and corneal opacification and vascularization due to a limbal stem cell deficiency and several other ocular and systemic pathologies. An abnormal epidermal- (skin) like phenotype of the corneal epithelium is associated with severe ocular surface disease in congenital aniridia. These changes in the corneal epithelium are associated with *PAX6* downregulation [5].

PAX6 is a transcription factor and has been shown to physically interact with various transcription factors such as c-Maf (Transcription factor Maf), Sox2 (transcription factor SOX-2), TFIID (transcription factor II D), pRb (retinoblastoma-associated protein), and MitF (microphthalmia-associated transcription factor) [6]. A physical or functional relationship between PAX6 and at-Ra has been investigated in the regulation of crystalline genes. The multifactorial enhancer elements in crystalline promotor regions have binding sites for PAX6 and RARs, which bind cooperatively and regulate gene expression [7]. In *Pax6* knockout mice, surface ectoderm-derived structures are unable to respond to retinoic acid after embryonic stage E9. Therefore, PAX6 is important for responding to at-RA in surface ectoderm-derived structures [8].

We have evidence that retinoic acid signaling may also be impaired in aniridia-associated keratopathy. The regulation of the transcriptional genes involved in the enzymatic conversion of retinoids as well as in retinoid- and fatty acid signaling is different in aniridia patient-derived ocular surface cells. These genes include *ADH7* (all-trans-retinol dehydrogenase 7), *ALDH1A1* (retinal dehydrogenase 1), *ALDH3A1* (aldehyde dehydrogenase, dimeric NADP-preferring), *PPARG* (peroxisome proliferator-activated receptor gamma), *FABP5* (fatty acid-binding protein 5), *CYP1B1* (cytochrome P450 1B1), *CYP26A1* (cytochrome P450 26A1), *STRA6* (receptor for retinol uptake STRA6), and *RDH10* (retinol dehydrogenase 10) [9,10].) The complex interactions between PAX6 and various transcription factors, including retinoid acid signaling, makes it difficult to interpret the expression profiles from aniridia subjects in order to identify the pathophysiological principles of the phenotypic changes of the corneal epithelium and the subsequent corneal opacification. The expression changes observed in patients could have different explanations. PAX6 might control the genes involved in retinoid metabolism directly, thereby changing the concentrations of active compounds such as at-RA in the cells. Additionally, PAX6 may possibly synergistically act with retinoic acid receptors during gene expression. This might also impact the genes needed for retinoid metabolism.

By manipulating genes at-RA levels, we sought to identify the genes that are related to the differentiation processes that are under the control of retinoic acid receptors or that are partly regulated by retinoic acid receptors.

Since the effect of retinoids at the transcription level has not been studied on limbal epithelial cells so far, we provided the first profile of a set of transcripts to generate an idea as to what extent retinoic acid metabolism and signaling could influence AAK-related expression profiles or which transcripts are sensitive to at-RA treatment.

Limitations: The experimental design of this study was not able to gain mechanistic insight into how PAX6 and retinoic acid directly interact with each other and how these effect the expression of their downstream genes. While working with primary cells with limited availability and low dividing capacity, it was also not possible to observe putative, short-term expression changes or PAX6 regulation in this setting.

## 2. Materials and Methods

Ethical considerations: The use of patient tissue in this study was approved by the Ethical Committee of Saarland/Germany (no. 226/15).

### 2.1. Cell Cultures and Treatment

Limbal tissue of healthy corneal donors was punched out using a 1.5 mm biopsy punch. The tissue was incubated with 500μL collagenase A (Roche Pharma AG, Basel, Switzerland) for 24 h at 37 °C. Primary limbal epithelial cell (LEC) clusters were filtered and were dissolved with 2.5 mL trypsin-EDTA-solution (Sigma-Aldrich GmbH, Deisenheim, Germany). Then, LEC were resuspended in 500 μL keratinocyte serum free medium (KSFM; Gibco, Carlsbad, CA, USA) and were seeded in one well of a 24-well plate. U LEC were incubated at 37 °C until confluence was reached. Thereafter, to transfer the LEC into a 6-well-plate, the cells were trypsinated and were dissolved in 3 mL of KSFM (Gibco, Carlsbad, CA, USA). The cells were passaged three times.

After growing to confluence in four 6-wells, each individual LEC culture was treated for 24 h with 0.5 µM or 1.5 µM retinol (Ret) or all-trans-retinoic acid (at-RA), respectively. The concentration used for the 48 h treatment was 1 µM or 5 µM retinol (Ret) or all-trans-retinoic acid (at-RA) (Sigma-Aldrich GmbH). Moreover, the LECs were treated for 24 h or 48 h with 0.5 µM, 1.0 µM, or 1.5 µM of RAR antagonists (AGN 193109) or 0.5 µM, 1.0 µM, or 1.5 µM of RXR antagonists (UVI 3003) (Sigma-Aldrich GmbH). Ret, at-RA, AGN 193109, and UVI 3003 were dissolved in dimethylsulfoxid (DMSO) and were diluted in KSFM to reach the above concentrations. The control cultures were treated using DMSO during the same incubation periods. Each treatment and each incubation time (24 h/48 h; Ret, at-RA, AGN, and UVI) was repeated three times with individual limbal epithelial cultures from different donors. The incubation happened under the same conditions as the cell culture procedures described above.

### 2.2. XTT Assay

Cell viability was evaluated using the XTT assay as follows: human LECs were seeded into 96-well cell culture plates in KSFM. After reaching 70% confluence, the culture medium was changed to a culture medium containing either 0.1 µM, 0.5 µM, 1 µM, or 5 µM retinol (Ret), all trans-retinoic acid (at-RA), AGN 193109 (AGN), or UVI 3003 (UVI) for 24 h and 48 h, respectively. Then, freshly prepared XTT solution was added to each well. As a negative control, XTT solution was added to a well without cells. After an incubation period of 30–60 min, absorbance was measured at 550 nm using a 96-well microplate reader (TECAN Infinite F50).

### 2.3. RNA/Protein Extraction and cDNA-Synthesis

RNA and protein were extracted using the RNA/DNA/Protein Purification Plus Micro Kit (Norgen, Thororld, ON, Canada) following the standardized protocol. The RNA-concentration was measured by means of UV/VIS-spectrophotometry (NanoDrop 1000 Spectrophotometer, Thermo Fisher Scientific, Waltham, MA, USA). For protein concentration measurements, a standardized Bradford test (Sigma-Aldrich GmbH) was conducted. The cDNA-synthesis was performed using the OneTaq RT-PCR Kit (New England BioLabs, Frankfurt a.M., Germany). An amount of 500 ng of RNA was used for one reaction (50 µL of the cDNA-synthesis).

### 2.4. Quantitative Polymerase Chain Reaction (qPCR)

For the qPCR, 1.25 µL Primer (Table A1 Appendix A) (QIAGEN GmbH, Hilden, Deutschland), 5µL AceQ SYBR qPCR Master Mix (Vazyme Biotech, Nanjing, China), and 0.5 µL cDNA were pipetted into one well of a 96-well plate according to the AceQ SYBR qPCR Kit (Vazyme Biotech, Nanjing, China). Beta-glucuronidase (GUSB) and TATA-binding-protein (TBP) were used as reference genes. From the Ct value and the calculated ΔCt and ΔΔCt values, the fold change (2^ΔΔCt^) was calculated. The following steps were repeated 40 times: initial denaturation to 95 °C for 5 min, denaturation at 95 °C for 10 s, and primer hybridization at 60° for 30 s. Elongation occurred when the sample was being heated up for the denaturation phases. The software QuantStudio™ Design & Analysis (Thermo Fischer, Waltham, MA, USA) was used for evaluation.

### 2.5. PAX6 Immunfluorescence Staining

To measure putative PAX6 protein expression changes, the LECs of three different donors were seeded into 96-well cell culture plates, and the cells were grown to 70% confluence. Individual wells were treated with DMSO (Ctrl) or with 0.1 µM, 1 µM, 1.5 µM, or 5 µM retinol (Ret), all trans-retinoic acid (at-RA), AGN 193109 (AGN), or UVI 3003 (UVI) for 24 h and 48 h, respectively.

During each step, the treatments were mixed in a total volume of 50 µL per well. After reaching confluence, the cells were directly fixed in 96-wells with 4% paraformaldehyde (PFA; PZN 2653025, Fischar). Cells were permeabilized with ice cold pure methanol for 5 min. Nonspecific epitopes were blocked with Phospahte buffered saline (PBS) containing 5% fetal calf serum (FCS) and 0.3% Triton-X-100 for 2 h. Primary antibodies were diluted in PBS containing 1% FCS and 0.1% Triton-X-100. For staining, mouse anti-PAX6 antibody (Santa Cruz, sc-32766, 1:50) was incubated overnight at 4 °C. Cells were washed three times with PBS containing 0.05% Tween. The secondary antibody, anti-mouse AF555 (Life Technologies, Cat. No. A21422), was diluted at a ration of 1:200 in washing buffer. After 1.5 h of incubation at room temperature, the cells were washed three times again, and the nuclei were stained with DAPI (Sigma, Cat. No. D9542). Cells were kept in PBS until image aquisition. Images were acquired with a Nikon Eclipse 100 microscope (Nikon, Japan) equipped with a 20× objective and suitable filter sets (DAPI, DAPI; Alexa 555, Cy3). For each treatment, two images were processed with ImageJ software. To compare the control LECs and treated cells, exposure times and image processing (e.g., brightness adjustments) were performed identically. For semi-quantitative analysis, nuclei were identified using DAPI staining, and ROIs were defined with ImageJ Software; then, the mean integrated grey values for each cell were measured in the PAX6 (Cy3) channel. The total PAX6 intensity of the controls was set to 100%, and the signals were normalized to this value, and these were compared to those in the treated cells.

### 2.6. Statistical Analysis

For all of the treatment types (Ret, at-RA, AGN, UVI), viability and mRNA expression values (ΔΔCt values) were compared to the controls using one-way ANOVA followed by Dunnett’s multiple comparisons test. Analysis was performed with GraphPad Prism 7.04 software (SanDiego, CA, USA).

## 3. Results

In the primary limbal epithelial cell cultures in vitro, the highest proportion of the cells belonged to transient amplifying cells. In vivo, at the corneal surface, these cells would further differentiate into mature corneal epithelial cells. This differentiation process is limited in cell culture, but could be triggered by air-lift culture, etc. Nevertheless, changes in differentiation marker expression are already measurable in LEC cultures, and these can be useful in identifying factors influencing differentiation at the gene expression level. As a first step, we started to investigate the possible adverse effects of different treatment options on LECs.

At-RA treatment for 24 h led to a minor but significant increase in viability at all concentrations, whereas 5 µM UVI led to a dramatic, significant decrease of cell viability. The same was true for the 48 h treatment using 5 µM AGN or UVI. There was no significant effect resulting from any of the other treatments on LEC viability, which was measured with the XTT assay (Figure 1).

MKI67 was used as a cell proliferation marker. *MKI76* mRNA expression is the highest in the M Phase of the cells [11,12]; therefore, it could also be used as a proliferation marker.

Retinol (Ret) had no significant impact on *MKI67* expression at any of the incubation times (Figure 2). At-RA treatment for 24 h also did not influence *MKI76* mRNA expression when using any of the concentrations. After at-RA treatment for 48 h, there was a significant decrease in *MKI67* mRNA expression when using the 1 µM (FC = 0.46) and 5 µM concentrations (FC = 0.001). AGN treatment for 24 h and 48 h led to significant *MKI67* downregulation at the 1.5 µM concentration (FC < 0.2), and there was a less pronounced downregulation at lower concentrations. Nevertheless, 1.0 µM AGN treatment for 48 h showed no significant fold change reduction. UVI treatment for 24 h evoked significant *MKI67*downregulation when using the 1.0 µM (FC = 0.49) and 1.5 µM concentrations (FC = 0.16).

As the aim of our study was to separate the response of at-RA from previously known PAX6-evoked expression changes, *PAX6* mRNA and protein expression measurements after at-RA and Ret treatment was necessary. The 24 h Ret treatment showed a significant but minor increase at the *PAX6* mRNA level when using the 0.5 µM (FC = 1.2) and 1.5 µM Ret concentrations (FC = 1.2). There was a significant PAX6 mRNA downregulation when using the 1 µM (FC = 0.7) and 5 µM (FC = 0.6) concentrations in the at-RA treatment group after 48 h but not after 24 h, whereas neither AGN and UVI treatment nor any other conditions changed *PAX6* mRNA expression (Figure 3A). In contrast, no significant difference in nuclear PAX6 staining was observed at any time points or concentrations compared to the controls (Figure 3B).

DSG1 (Desmoglein-1) is a cellular junction protein, and it has been reported to be downregulated in Sey^+/−^ (Pax6^+/−^) mice, which is an animal model for congenital aniridia. DSG1 expression was also downregulated in anirida patient-derived ocular surface cells when the siRNA-based PAX6 knockdown cell model was used [10,13]. SPINK7 is a protease inhibitor of kallikrein serine proteases (KLKs), which are also able to cleave DSG1, for example [14]. SPINK7 was found to be downregulated in our siRNA-based aniridia cell model (artificial PAX6 knockdown in LECs) as well [10]. It remains unclear how these genes are regulated in limbal epithelial cells.

*DSG1* mRNA expression tended to result in a reduction after 24 h Ret incubation, and *DSG1* expression was significantly reduced after 48 h when using the 1 µM (FC = 0.06) or 5 µM (FC = 0.06) Ret concentrations. The treatments using the 1 µM or 5 µM at-RA concentrations also led to reductions in *DSG1* (FC = 0.07 and FC = 0.3). In contrast, after treatment with the RAR antagonist AGN, *DSG1* was upregulated after 24 h at all of the tested concentrations (0.5 µM (FC = 2.1); 1.0 µM (FC = 3.5); 1.5 µM (FC = 3.2)). The upregulation after the 48 h AGN incubation was not significant. *DSG1* mRNA expression did not change significantly upon UVI treatment (Figure 4). *SPINK7* mRNA expression did not decrease after the 24 h Ret treatment but decreased significantly after the 48 h Ret treatment with the 1 µM (FC = 0.3) or 5µM concentrations (FC = 0.3). The 24 h incubation with at-RA led to significant *SPINK7* downregulation when using the 0.5 µM (FC = 0.52) and 1.5 µM concentrations (FC = 0.52) and to an even more prominent *SPINK7* reduction after the 48 h at-RA treatment with the 1 µM (FC = 0.2) or 5 µM concentrations (FC = 0.1). AGN and UVI treatment had no significant impact on *SPINK7* expression (Figure 4).

Keratins are important differentiation markers of the ocular surface. Their relative expression in LECs compared to differentiated corneal epithelial cells is very low. KRT3 and KRT12 markers are downregulated in ocular surface of subjects with aniridia and are described as being regulated by *PAX6* in vitro as well [15,16]. However, further signals may be needed to better specify corneal cell identity. KRT19 is regarded as a conjunctival marker.

*KRT3* mRNA expression was downregulated after the 48 h Ret treatment using the 1.0 µM (FC = 0.47) and 1.5 µM concentrations (FC = 0.49) but not after the 24 h incubation time (Figure 5). The 24 h Ret treatment showed an upregulation trend instead. There were no significant expression changes when using the at-RA treatment for 24 h or for 48 h. AGN and UVI did not show a significant effect on *KRT3* expression. *KRT12* was significantly upregulated after the 24 h Ret treatment using the 1.5 µM concentration (FC = 1.4) and after 48 h with the 1 µM Ret treatment (FC = 1.7). A similar pattern was observed after the at-RA treatment, with a *KRT12* upregulation occurring when the 1.5 µM concentration (FC = 1.7) was used after 24 h (Figure 5). After the 48 h at-RA treatment, a *KRT12* upregulation at the 1 µM concentration (FC = 2.8) was observed. AGN treatment for 24 h (FC = 0.5–0.7) and 48 h (FC = 0.3–0.7) led to a significant *KRT12* downregulation when using all of the concentrations. UVI treatment did not lead to significant changes.

*KRT19* mRNA expression was only significantly upregulated after the 48 h at-RA treatment when using the 5 µM concentration (FC = 2.9). *KRT19* downregulation following AGN treatment was observed after 24 h when using the 0.5 µM (FC = 0.6) and 1.0 µM AGN concentrations (FC = 0.7).

Enzymes and components of retinol and fatty acid metabolism (ADH7, ALDH1A1, RBP1, RDH10, CRABP2, ELOVL7, FABP5, and PPARG) were altered on the ocular surface cells from aniridia patients and in the siRNA-based aniridia cell model. Our purpose was to monitor the gene expression changes of these genes in response to at-RA and pan-RAR and pan-RXR antagonists. This should allow us to draw conclusions regarding the observed expression patterns in the cells of aniridia subjects and in aniridia cellular models are affected by retinoid metabolism and which pathways are likely involved. Several transcripts were significantly altered upon treatment: See the results of transcripts ADH7, ALDH1A1, RBP1, RDH10, and CRABP2 (Figure A1) and the results of ELOVL7, FABP5, and PPARG (Figure A2). However, no clear conclusions/metabolic linkages could be drawn from these transcriptional results (See discussion for further interpretation and limitations).

## 4. Discussion

### 4.1. Effect of Retinol and Retinoic Acid on MKI67 Expression

The effect of retinol and at-RA on differentiation and proliferation has been studied by several research groups previously, who have analysed limbal epithelial cells and skin epidermal cells [17,18,19]. The 1 µM at-RA dosage used in this study has been reported to prevent abnormal differentiation and to reduce colony forming efficiency (measures of high proliferative stem cells) [20]. The highly significant downregulation of *MKI67* mRNA expression (Figure 2) is in line with the decrease of colony forming efficiency in limbal epithelial cells upon at-RA treatment, as reported by others.

### 4.2. Effect of Retinol and Retinoic Acid on LEC

Retinol evoked similar effects on limbal epithelial cell mRNA expression to those of at-RA treatment for some of the transcripts in our present study. This could be explained by the simple conversion of retinol to at-RA in corneal epithelial cells. For some experiments (DSG1), there was a weaker or no response after 24 h compared to after the 48 h Ret treatment (DSG1, SPINK7 and KRT3), which could partly be explained by the fact that Ret first needs to be converted to at-RA in the LECs. The presence of all of the enzymatic components converting retinol to active at-RA in corneal epithelial cells and the enzymatic conversion of retinol to at-RA has been previously observed in corneal tissue [21].

### 4.3. Similarities and Differences in RA Signaling and PAX6 Haploinsufficiency

PAX6 and retinoid signaling have already been linked previously, but these were mostly discussed and observed in the developmental processes of the eye and brain [8,22]. In neurons, PAX6 is reported to be upregulated by at-RA, although the *PAX6* promotor does not contain functional retinoic acid response elements [22]. In *PAX6^−/−^* rats, *ALDH1A3* is not expressed, and it has been speculated that *ALDH1A3 (RALDH-3*) is a downstream target of *PAX6* [23].

In prior experiments, we observed an enrichment of deregulated genes that are associated with retinoid metabolism in PAX6^+/-^ epithelial cells that had been derived from patients and in cellular models. For example *ADH7*, *ALDH1A1*, and *ALDHA1A3* are downregulated in PAX6^+/−^-derived patient cells [9,10]. Since ADH7 can oxidize retinol to its aldehyde form [24,25] and since ALDH1A1/3 are enzymes that can oxidize retinol to retinoic acid [26], we speculated, that a part of the expression changes that have been observed in cellular models and in patient cells are related to a deficient at-RA metabolism. Our hypothesis was that a locally reduced at-RA concentration may lead to an impaired differentiation of primary LEC.

In addition to the changes in RA-associated transcripts, differentiation makers of the corneal epithelium such as *DSG1* are downregulated in aniridia mouse models and upon PAX6 knockdown in primary corneal epithelial cells [13,27].

In order to make sure that the effect of at-RA on gene expression is not through the feedack of at-RA on PAX6 expression, we analysed to what extent at-RA influences PAX6 expression. The treatment of primary limbal epithelial cells with Ret for 24 h provoked a slight increase of *PAX6* mRNA expression, and 48 h of at-RA treatment was related to a downregulation of *PAX6* mRNA. With the immunfluoresence imaging of nuclear PAX6 staining, we did not detect an effect of Ret treatment on the PAX6 protein level under any treatment condition (Figure 3B). One should consider that we did not determine any dynamic changes in the PAX6 protein level over time. Since posttranslational regulation or narrow time scaling could not be included in our experimental setup, a small amount of uncertainty regarding a putative influence of RA on PAX6 protein level remains.

Therefore, to the best of our knowledge, the administration of Ret and at-RA to the LECs of healthy donors could influence differentiation marker expression that is independent of *PAX6*. At-RA decreased *SPINK7* and *DSG1* mRNA expression (Figure 4), supporting evidence that some expression changes observed in aniridia patients are likely due to the overactivation of retinoic acid signaling or the accumulation of at-RA. These expression changes were similar to the changes observed in aniridia corneal epithelial cells and upon the PAX6 siRNA knockdown of LEC (See Table 1).

*DSG1* expression is a very stable readout for *PAX6* defienceny conditions in differentiating ocular surface cells. *DSG1* is downregulated in *Sey^+/−^* (*PAX6*^+/*−*^) mice [13], in primary corneal epithelial cells upon complete *PAX6* knockout [27], in ocular surface epithelial cells of human anridia patients, and upon the siRNA knockdown of the primary limbal epithelial cells of healthy donors [10].The data from the present study indicate that *DSG1* downregulation happens due to a disturbed retinol metabolism and that DSG1 is not directly regulated by PAX6. The evidence for this would be as follows:

Treatment with retinol as well as with at-RA leads to a significant and sustained downregulation of *DSG1* mRNA. Admission of the AGN antagonist (RAR receptor antagonist) led to a *DSG1* mRNA expression increase (Figure 4). This observation was similar in skin epidermal cells, as at-RA led to the downregulation of DSG1 mRNA and protein expression [28].

A similar argumentation could be also true for SPINK7. SPINK7 was identified to be potentially regulated by *PAX6* in our own previous screening [10]. It may inhibit Kallikrein serine proteases (KLKs), which are able to cleave, e.g., DSG1 [14]. Therefore, a functional or regulatory relationship to *DSG1* is conceivable. *SPINK7* was downregulated in the samples stimulated with Ret and at-RA, but no clear upregulation by AGN antagonist was observed. SPINK7 may be affected due to the changes in at-RA signaling rather than being a direct downstream target of PAX6.

The underlying mechanisms of the regulation of keratins could not be presented as clearly. Several keratins are well-described selective differentiation markers for the corneal epithelium. Their altered expression pattern upon severe ocular surface disease includes the loss of KRT12 and KRT3 expression [5]. In contrast to the other investigated markers in this study, corneal keratins have been linked to PAX6 expression by several other groups [16]. PAX6 protein expression is necessary for KRT12 mRNA and protein expression, but PAX6 protein alone is not sufficient for its expression, as shown by promotor analysis [15,29]. CRISPR/CAS-mediated PAX6 knockout in corneal epithelial cells resulted in strong KRT12 and KRT3 downregulation [27].

Analysing in vitro or in vivo, several keratins have diverse responses to at-RA treatment in case of the skin [30]. Although keratin expression is influenced by at-RA, the RAR/RXR receptors are not involved in the regulation of all keratins since the retinoid response elements (RAREs) cannot be found in the keratin promotor region of KRT2 and KRT4, for example [30].

Only a few of the tested conditions in our study showed the upregulation of KRT12 and the downregulation of KRT3 in the transcriptional data. However, Kim et al. described KRT12 protein upregulation and KRT3 protein downregulation using a 1 µM at-RA treatment in stratified corneal epithelial sheets [17]. The alterations in keratin mRNA expression, which were also described in our study, match these previous observations [17]. Since our results are not clear concerning the influence of at-RA on KRT expression, this may point to a limitation of our study: it is impossible to extrapolate the observed but minor expression changes to biological means during differentation processes.

An upregulation of *KRT19* mRNA expression upon at-RA stimulation has been reported in several studies, especially those analysing skin epidermal cells in vitro and in vivo. In LECs, in our present study, only the 48 h at-RA stimulation with a high concentration evoked an upregulation. It has been shown by others that *KRT19* mRNA elevation is not a consequence of enhanced mRNA transcription but instead happens due to the posttranslational stabilization of the transcript [31]. Significant *KRT19* mRNA downregulation (FC > 0.6) after 24 h with RAR antagonist treatment may be due to revertion of the unknown mechanism of mRNA stabilization.

In conclusion, a minor impact of at-RA treatment on keratin expression can be observed in LECs. The role of at-RA in the regulation of keratins and its role in AAK pathogenesis cannot be be drawn from in vitro LEC data in this study.

### 4.4. RA Transport, Metabolism and Signaling/Transcripts Involved in Fatty Acid Metabolism and/or Signaling

In contrast to the relative clear changes in DSG1, SPINK7, or keratin expression patterns, ADH7, ALDH1A1, RBP1, RDH10, and CRABP2 expression (Figure A1) or ELOVL7, FABP5, and PPARG expression (Figure A2) did not show clear trends as a response to Ret or at-RA treatment. The responses were often restricted to one concentration of the experimental series. In addition, the response to the antagonists was not specific; therefore, we were unable to assign the response to a retinoic acid receptor class. This might be due to following reasons or experimental limitations:

When applying higher concentrations of at-RA, there is also a risk for activating cis-RA-related signaling, as cis-RA is always present as an unavoidable contamination in at-RA solutions and could lead to over-activation of the RXR-RAR heterodimer [32]. In addition, UVI can transactivate PPARG at higher concentrations [33].

Extensive titration of the concentations helps to avoid cross activity and supports the use of more specific agonists and antagonists. However, there is then a demand for an extensive use of primary or other cells. As these are not available, other tools beyond drug treatment should be prefered in the future to study the impact of at-RA signaling on gene expression in limbal epithelial cells.

Expression of retinoic acid binding proteins CRABP2, RBP1, and retinol dehydrogenase RDH10 may change as a response to altered retinoid levels or may be influenced by activated RAR and RXR receptors.

*ADH7* mRNA expression, in contrast to the siRNA experiments reported by us, was not significantly regulated when at-RA was administered alone, but followed the same trend for some concentrations after 48 h with the 1 µM Ret or 1 µM at-RA treatment. Nevertheless, when using a higher Ret or at-RA dosage, the changes were even less pronounced (Figure A1). RAR and RXR activation both might suppress ADH7 expression, but a transactivation of PPARG through the used high antagonist concentrations could also interfere with the regulation of AHD7.

ADH7 enzymatic activity is not only restricted to retinoids but is also restricted to medium chain fatty acids (MCFs). Therefore, we also checked the deregulated transcripts in the fatty acid metabolism.

Changes in FABP5, PPARG, and ELOVL7 transcription, which are often observed in anirida, could be explained through the crosstalk of fatty acids with retinoic acid signaling. Responses of these latter-mentioned transcripts to at-RA could give valuable hints for such a crosstalk. FABP5 and PPARG may both effect at-RA and fatty acid (FA) signaling [34,35], and ELOVL7 is related to fatty acid metabolism [36]. ELOVL7 and PPARG were significantly upregulated after the 24 h und 48 h AGN treatments. After the UVI treatment, gene expression tended to be similar. The regulation of these transcripts could be influenced by other sensors that are able to recognise the lipophilic molecular structure of the molecules that were used for treatment in the present study rather than RAR/RXR receptors, which were the actual target of the treatment. Therefore, the observed ELOVL7, FABP5, and PPARG transcriptional changes in aniridia patient cells cannot be based on a simple change in the at-RA metabolism but are related to more complex changes in cellular signaling pathways.

The functional balance between at-RA and fatty acid metabolism and signaling could be important to maintain an ocular surface epithelial phenotype without keratinization and cornified envelope formation. However, such a connection to at-RA signaling could not be established in our in vitro systems, which only anlaysed the short-term impact of at-RA concentration changes, although the *ELOVL7*, *FABP5*, and *PPARG* transcripts responded to at-RA and AGN.

Summarizing all of the described data, we conclude that the influence of Ret or at-RA on gene expression in aniridia LECs exists independently from direct changes in PAX6 mRNA or protein expression. Our in vitro data suggest that an elevated Ret or at-RA level correlates with reduced *DSG1* and *SPINK7* mRNA expression, and some concentrations probablyt evoke reduced *ADH7* expression. In the corneal and conjunctival epithelial cells of PAX6^+/-^ patients, DSG1, ADH7, and probably SPINK7 downregulation could be evoked by elevated Ret or at-RA levels. A more global regulation of at-RA signaling that could interfere with changes in PAX6 expression is likely involved but is not adressed in the presented data.

At-RA signaling, which influences tissue-specific controled genes, is not understood in detail. The binding of RAR and RXR to specific promotor regions and associated chromatin remodeling as well as crosstalk to other signals such as FGF/MAPK, TFGβ/BMP, hedgehog, Notch Wnt, and Jak/STAT pathways are reported in other systems that are already separate from the eye [37]. Therefore, a more detailed proteomic and metabolic data analysis obtained from patient epithelial cells is necessary in the future.

## 5. Conclusions

Despite the limitations of our in vitro systems and the restrictions for transcriptional analysis, we were able to provide supporting data that retinoic acid signaling impacts several transcripts that are markers for corneal epithelial differentiation. These include *DSG1*, *SPINK7*, and to lesser extent, the Keratins *KRT12* and *KRT3*.

## Figures and Tables

**Figure 1 biomolecules-11-01651-f001:**
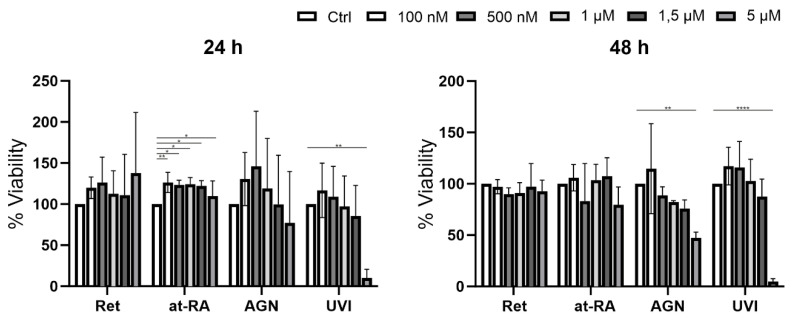
XTT viability assay of primary LEC treated with retinol (Ret), all trans-retinoic acid (at-RA), AGN 193109 (AGN), or UVI 3003 (UVI) for 24 h and 48 h at increasing concentrations (controls without treatment) of 100 nM, 500 nM, 1 µM, 1.5 µM, 5 µM), respectively. * *p* ≤ 0.05; ** *p* ≤ 0.01; **** *p* ≤ 0.0001; *n* = 3.

**Figure 2 biomolecules-11-01651-f002:**
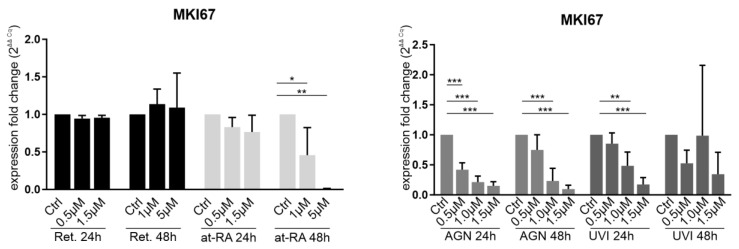
qRT-PCR analysis of the proliferation marker *MKI67* (Ki67) in primary limbal epithelial cells treated with retinol (Ret), all trans-retinoic acid (at-RA), AGN 193109 (AGN), or UVI 3003 (UVI) for 24 h and 48 h. Expression fold changes (FC) were calculated relative to each control with the ΔΔCt method. * *p* ≤ 0.05; ** *p* ≤ 0.01; *** *p* ≤ 0.001; *n* = 3.

**Figure 3 biomolecules-11-01651-f003:**
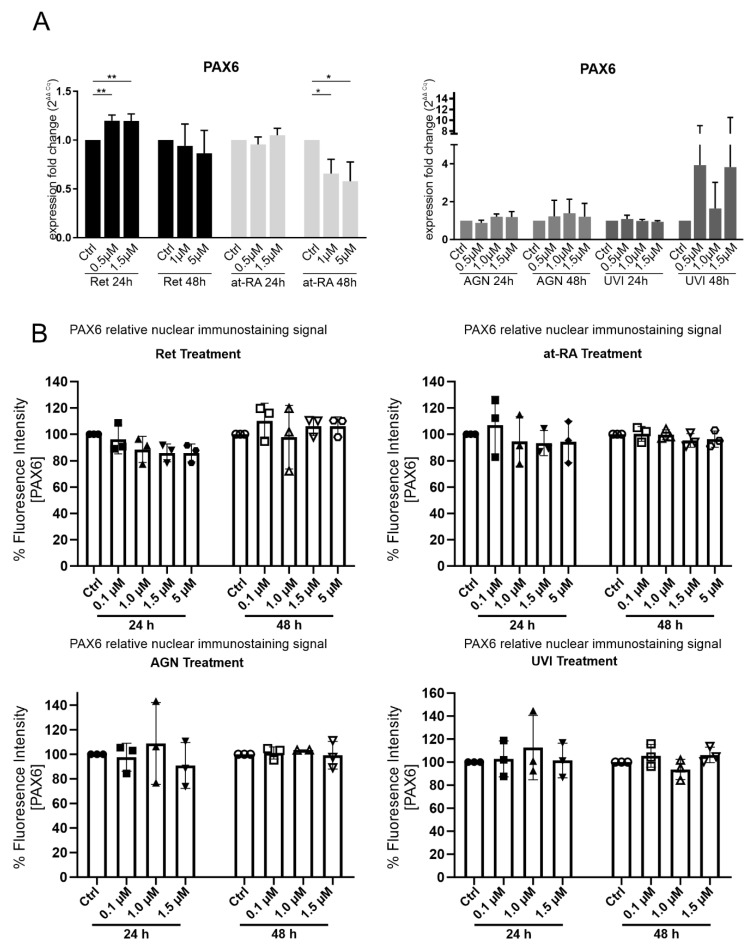
PAX6 mRNA (**A**) expression and PAX6-relative nuclear immunostaining signal (**B**) in primary limbal epithelial cells treated with retinol (Ret), all trans-retinoic acid (at-RA), AGN 193109 (AGN), or UVI 3003 (UVI), for 24 h and 48 h respectively. The used concentrations are indicated. (**A**) qRT-PCR analysis of PAX6 mRNA extracted from primary limbal epithelial cells (LEC). Expression fold changes (FC) are calculated relative to each control with the ΔΔCt method. (**B**) Nuclear PAX6 immunofluorescence staining of LEC cultures grown in 96 wells following the indicated treatment. The relative fluorescence signal of controls and treatment groups is represented as mean ± standard deviation (SD). Two images of every sample were analyzed and PAX6 nuclear intensity was normalized to control treatment of the same LEC preparation (set as 100%). * *p* ≤ 0.05; ** *p* ≤ 0.01; *n* = 3.

**Figure 4 biomolecules-11-01651-f004:**
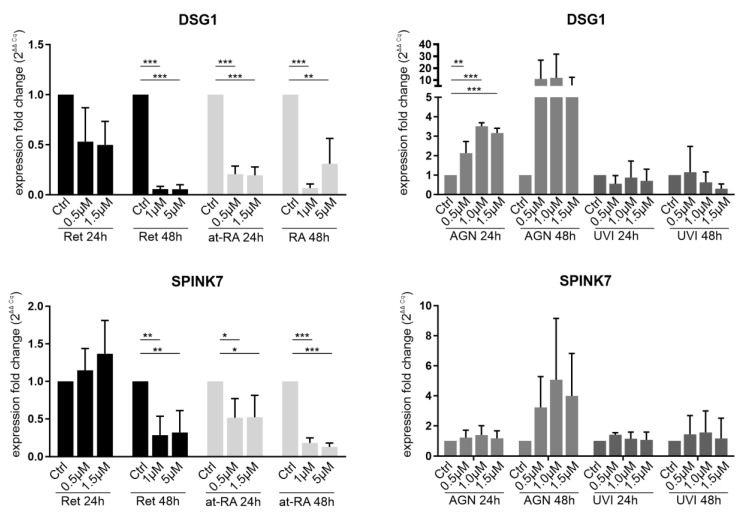
qRT-PCR analysis of DSG1 and SPINK7 transcripts in primary limbal epithelial cells (LEC) treated with retinol (Ret), all trans-retinoic acid (at-RA), AGN 193109 (AGN), or UVI 3003 (UVI) for 24 h and 48 h. The transcripts shown here are related to the PAX6^+/−^ phenotype in patients and cell models. Expression fold changes (FC) are calculated relative to each control using the ΔΔCt method. * *p* ≤ 0.05; ** *p* ≤ 0.01; *** *p* ≤ 0.001; *n* = 3.

**Figure 5 biomolecules-11-01651-f005:**
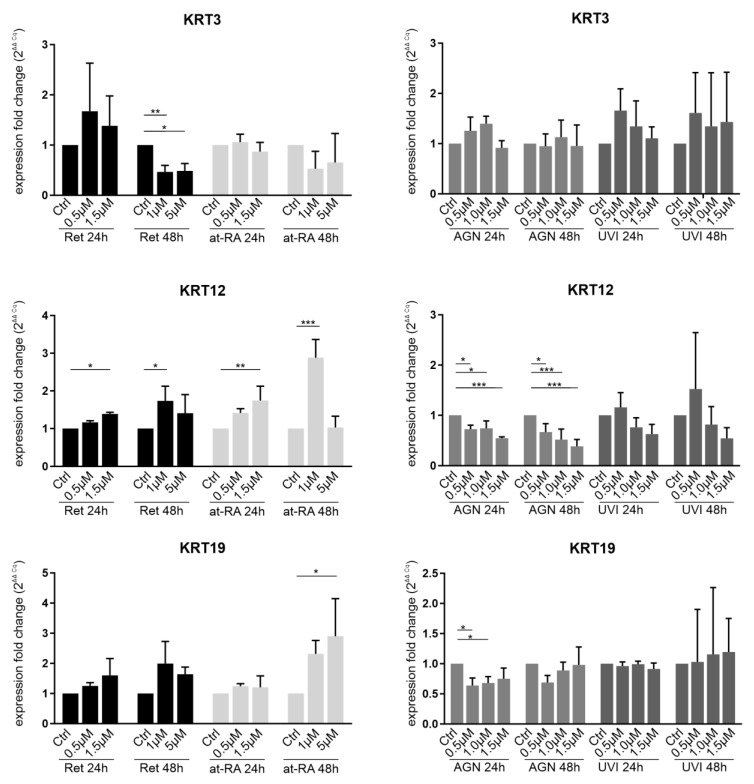
qRT-PCR analysis of KRT3, KRT12, and KRT19 transcripts in primary limbal epithelial cells (LEC) treated with retinol (Ret), all trans-retinoic acid (at-RA), AGN 193109 (AGN), or UVI 3003 (UVI) for 24 h and 48 h. The transcripts shown here are related to the corneal (KRT3/KRT12) and limbal/conjunctival epithelial phenotype (KRT19). Expression fold changes (FC) are calculated relative to each control using the ΔΔCt method. * *p* ≤ 0.05; ** *p* ≤ 0.01; *** *p* ≤ 0.001; *n* = 3.

**Table 1 biomolecules-11-01651-t001:** mRNA expression changes in limbal epithelial cells (LEC) following treatment and in aniridia patient cells. Transcripts with similar behaviour through retinol (Ret)/all trans-retinoic acid (at-RA) stimulation and in aniridia-related models or primary limbal epithelial cells (LEC) are highlighted in green. Transcripts with unexpected response in one of the cellular systems are highlighted in yellow. Transcripts not reacting according our hypothesis are not highlighted. RAR: retinoic acid receptor; siPAX6: siRNA based cell model with PAX6 knockdown; ^1^ = primary LECs from aniridia subjects; ^2^ = primary LECs from healthy subjects treated with siRNA against *PAX6*; “---” = no data available; * = unpublished Data.

Transcript	LEC + Ret/at-RA (Present Study)	LEC + RAR Antagonist (Present Study)	Aniridia (PAX6^+/−^) LEC ^1^ /LEC (si*PAX6*) ^2^ [10]	Aniridia (PAX6^+/−^) Conjunctiva [9]
*ADH7*	**DOWN (n.s)**	UP	**DOWN ^1^/DOWN** ^2^	**DOWN**
*DSG1*	**DOWN**	UP	**DOWN ^1^/DOWN** ^2^	**DOWN**
*SPINK7*	**DOWN**	No change	**DOWN ^1^/DOWN** ^2^	---
*ALDH1A1*	No change	No change	DOWN ^1^/DOWN ^2^	---
*KRT12*	UP	DOWN	DOWN ^1^/No change ^2^	---
*KRT3*	**DOWN**	UP	**DOWN**^1^/**DOWN**^2^	---
*KRT19*	UP	DOWN	--- ^1^/--- ^2^	---
*ELOVL7*	UP	UP	UP ^1^/No Change *	UP
*FABP5*	No Change	UP	**DOWN** ^1^/**DOWN** *	**DOWN**
*PPARG*	Unclear	UP	UP ^1^/DOWN *	UP
*CRABP2*	**DOWN**	UP	**DOWN** ^1^/**DOWN** *	UP
*RBP1*	UP	No change	No change ^1^/DOWN *	UP
*RDH10*	UP	UP	No change ^1^/DOWN *	UP

## Data Availability

Exclude this Statement.

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
