# Peer review of "Similarities in DSG1 and KRT3 Downregulation through Retinoic Acid Treatment and PAX6 Knockdown Related Expression Profiles: Does PAX6 Affect RA Signaling in Limbal Epithelial Cells?"

_biomolecules, 2021, doi:10.3390/biom11111651_

Round 1
Reviewer 1 Report
The manuscript does address issues regarding the potential link between the protein PAX6 and the retinoic acid signaling in limbal epithelial cells. This paper is a resubmission. Even the manuscript is a little improved and that the scientific question raised is interesting, and although the authors do not over-interpret the results and sometimes remain cautious in their conclusions or interpretations, the conclusions do not allow a mechanistic breakthrough and some aspects pointed out in the first submission remain. Overall, the text remains obscure and difficult to read (typos, English, turns of phrase, vocabulary). Abbreviations are not consistent. (e.g. for all-trans retinoic acid).
This article presents observations of experiments that appear to be technically correct, but whose design is flawed.
Among several concerns, one in particular affects the relevance and scope of the work done.
The choice of ligands used and their experimental conditions of use are questionable and do not allow to attribute their effects to their action on the RAR and RXR receptors, and thus to interpret the experimental results in a relevant manner.
Firstly, why do the authors use ligands at much higher concentrations than their Kds, whether for RAR or RXR. This is especially damaging for the retinoic acid and the synthetic antagonist ligands (AGN193109 for RAR and UVI3003 for RXR). The authors comment on this problem (line 410) and it is understandable that it is difficult to repeat all the analysis with adequate ligand conditions. But unfortunately this point strongly weakens the present article.
In solutions of at-RA, this molecule is often converted to its 9cis-RA isomer, which has a high affinity and agonist activity for RXR. At 1µM and above of at-RA, the amount of 9cis-RA is often sufficient to activate RXR. As RXR is the heterodimerisation partner of many other nuclear receptors, other signaling pathways may be involved. To be sure that the observed effects can be attributed to RAR, at-RA should be used at a maximum of 0.1µM, or alternatively a synthetic agonist such as TTNPB. If under such conditions an effect of the molecule is not observed, then one cannot conclude that RAR is involved.
Therefore, the choice to use antagonists alone, and not in competition with agonists, is surprising. Under these conditions, in order to observe effects, it would be necessary for the receptors to be already activated by an endogenous molecule or for the antagonist to exert an effect by itself.
AGN is a particular RAR antagonist because this compound is an inverse agonist and induces an increase of the transcriptional repression through RAR. The effects observed with this molecule in the present study may be attributed either to this property or to the activation of another pathway given the concentration used. A competition of the effects exerted by at-RAR with AGN would allow better targeting of RAR.
Concerning the RXR antagonist UVI3003, this molecule is inactive on RXR. Therefore, if an effect is observed, RXR must already be activated by an endogenous ligand whose existence has not been demonstrated. At concentrations higher than 1µM, UVI can activate the PPARG. Yet the authors conclude that RXR is involved, for example line 423: “A significant upregulation was observed after 24 h AGN and UVI treatment with high concentrations, which means that RAR and RXR activation both might suppress ADH7 expression.”
Reviewer 2 Report
The manuscript entitled “Similarities in DSG1 and KRT3 downregulation through retinoic acid treatment and PAX6 knockdown: Does PAX6 affect RA signaling in limbal epithelial cells?” by Latta et al. describes the results obtained in experiments in which human limbal epithelial cells (LEC) to exposed to four compounds related to vitamin A and vitamin A metabolites. The paper has been substantially improved in comparison with previously submitted version. However, still some aspects should be corrected before publication. These are:
- Language.
- line 85: “manipulating at-RA level genes” should be phrased “manipulating the genes responsible for keeping at-RA levels”.
- line 96 insert “While” before “working”.
- lines 144-145: Sentence “From the Ct value and the calculated ΔCt and ΔΔCt values, the fold change (2ΔΔCt) was calculated.” Should be moved after “Elongation occurred during heating up for denaturation.”
- line 400: correct “singaling” to “signalling”.
- line 472: replace “overtaken” with “translated”.
- Discussion in sections in sections 4.1. and 4.2. is well written and easy to understand. Remaining sections of the discussion must be rewritten, because it is really hard to understand them.
Author Response
Reviewer #1
The manuscript entitled “Similarities in DSG1 and KRT3 downregulation through retinoic acid treatment and PAX6 knockdown: Does PAX6 affect RA signaling in limbal epithelial cells?” by Latta et al. describes the results obtained in experiments in which human limbal epithelial cells (LEC) to exposed to four compounds related to vitamin A and vitamin A metabolites. The paper has been substantially improved in comparison with previously submitted version. However, still some aspects should be corrected before publication. These are:
- Language.
- line 85: “manipulating at-RA level genes” should be phrased “manipulating the genes responsible for keeping at-RA levels”.
- line 96 insert “While” before “working”.
- lines 144-145: Sentence “From the Ct value and the calculated ΔCt and ΔΔCt values, the fold change (2ΔΔCt) was calculated.” Should be moved after “Elongation occurred during heating up for denaturation.”
- line 400: correct “singaling” to “signalling”.
- line 472: replace “overtaken” with “translated”.
We thank reviewer #1 for these comments. We have corrected all mentioned errors and have checked the entire manuscript again for linguistic and spelling errors.
- Discussion in sections in sections 4.1. and 4.2. is well written and easy to understand. Remaining sections of the discussion must be rewritten, because it is really hard to understand them.
We thank reviewer #1 for this useful advice. In accordance with Reviewer#2 we have thoroughly revised the discussion and omitted unnecessary details. We hope that it is now much more understandable.
Reviewer 3 Report
The manuscript "Similarities in DSG1 and KRT3 downregulation through retinoic acid treatment and PAX6 knockdown related expression profiles: Does PAX6 affect RA signaling in limbal epithelial cells?" written by Latta L. and coauthors analyses the biology of primary limbal epithelial cells treated with retinoic acid and its derivate, as well as with antagonists of retinoic acid receptors. The expression of a number of differentiation markers was analyzed and data compared with those obtained from cells with PAX6 downregulation linked with aniridia.
The authors accepted the comments previously written and corrected the manuscript. In general, the manuscript is well written, except Discussion which is still unnecessary too long and could be better organized.
At the beginning of the Results, there could be a small introduction to present the cells, aims of the treatment, activity of the inhibitors. Figures (S1 and S2) of qPCR of different genes analyzed could be inserted in the Results.
Discussion could begin with proliferation and viability analysis, and then discuss the gene expression. Several paragraphs, when having the same topic can be fused to one. The paragraph 445-452 seems to summarize the data so the following paragraphs should be put in front of it. In the Conclusions, it is enough to use the first paragraph.
Minor comments:
line 59: proteins are not written in italics.
There is no need of new paragraphs if the topic is the same (i. e. lines 79-84)
Units should be written separately from the numbers.
line 191: a proliferation marker
line 193, 196: MKI67
line 217: previously
line 219: PAX6 mRNA
line 234: KLK: explanation of the abbreviation
line 410: cross activity
Author Response
Reviewer #2
The manuscript "Similarities in DSG1 and KRT3 downregulation through retinoic acid treatment and PAX6 knockdown related expression profiles: Does PAX6 affect RA signaling in limbal epithelial cells?" written by Latta L. and coauthors analyses the biology of primary limbal epithelial cells treated with retinoic acid and its derivate, as well as with antagonists of retinoic acid receptors. The expression of a number of differentiation markers was analyzed and data compared with those obtained from cells with PAX6 downregulation linked with aniridia.
The authors accepted the comments previously written and corrected the manuscript. In general, the manuscript is well written, except Discussion which is still unnecessary too long and could be better organized.
At the beginning of the Results, there could be a small introduction to present the cells, aims of the treatment, activity of the inhibitors. Figures (S1 and S2) of qPCR of different genes analyzed could be inserted in the Results.
We thank reviewer #2 for this advice. As suggested, we added a small introduction to the beginning of the Results, we hope that it helps to follow the ideas.
Nevertheless, we decided to keep S1 and S2 Data as Supplement, since these findings could distract the reader from the results, decreasing clarity.
Discussion could begin with proliferation and viability analysis, and then discuss the gene expression. Several paragraphs, when having the same topic can be fused to one. The paragraph 445-452 seems to summarize the data so the following paragraphs should be put in front of it. In the Conclusions, it is enough to use the first paragraph.
We thank reviewer #2 for this useful advice. In accordance with Reviewer#1 we have thoroughly revised the discussion and omitted unnecessary details. We hope it is now much more understandable.
Minor comments:
line 59: proteins are not written in italics.
There is no need of new paragraphs if the topic is the same (i. e. lines 79-84)
Units should be written separately from the numbers.
line 191: a proliferation marker
line 193, 196: MKI67
line 217: previously
line 219: PAX6 mRNA
line 234: KLK: explanation of the abbreviation
line 410: cross activity
We thank reviewer #2 for these comments. We have corrected the mentioned errors and have checked the entire manuscript again for linguistic and spelling errors.

Round 2
Reviewer 1 Report
In their response the authors argue as follows:
“In cell culture, it is more common to use IC50 / EC50 values, which differ in some cases dramatically from Kds (which are biochemically determined). These values refer to the different biological availability of the substance to the cells, in culture. This includes binding to proteins in growth media and penetration into cells etc. A practical use of at-RA in µM range in cell cultures has been presented by other studies already (Balmer & Blomhoff, 2002; Kim et al., 2012; Lee et al., 2009). In addition, by the given 24 h and 48 h incubation times, at-RA, Retinol, AGN and UVI added to the cell culture might degrade over time (especially in serum free media (Sharow et al., 2012).”
First synthetic retinoids are stable. On the other hand, on most cell culture experiments there is a correlation for the cellular activity of a retinoid between Kds and IC50 / EC50 values. Although small differences may sometimes exist due to certain reasons cited by the authors, there is no justification for using retinoids at such high concentrations. The case of at-RA is special as I have already explained (In solutions of at-RA, this molecule is often converted to its 9cis-RA isomer, which has a high affinity and agonist activity for RXR. As a result, in the context of the RXR-RAR heterodimer, when both partners are activated a synergy occurs which affects the apparent dose-response curve of at-RA. At 1µM and above of at-RA, the amount of 9cis-RA is often sufficient to activate RXR. As RXR is the heterodimerisation partner of many other nuclear receptors, other signaling pathways may be involved. To be sure that the observed effects can be attributed to RAR, at-RA should be used at a maximum of 0.1µM). This is why the use of a synthetic agonist such as TTNPB is recommended.
I reiterate my comment regarding the conditions of use of retinoids in this study. There are many examples in the literature of work with questionable findings due to the use of excessive doses of retinoids.
However, the authors correctly state the limitations of their study and this article represents a substantial amount of work and has been improved, and raises an interesting question. The latest version of the manuscript is acceptable.
This manuscript is a resubmission of an earlier submission. The following is a list of the peer review reports and author responses from that submission.
Round 1
Reviewer 1 Report
In this paper Latta et al. try to elucidate if PAX6 affects RA signaling in
limbal epithelial cells. Interestingly, for this study, they use human LEC isolated from healthy donor corneas.
However, few minor edits would improve the paper:
- Line 184- I would suggest specifying that all the treatments are at 24h. Next, in the discussion section, I would discuss why the 48h treatments are not included.
- AGN 193109 is a pan-retinoic acid receptor (RAR) antagonist and UVI 3003 is a highly selective antagonist of retinoid X receptor (RXR). Have the authors considered to include also a specific RXR agonist? (e.g bexarotene, HX531)?
- How they have ruled out that PAX6 mRNA expression in unchanged under all the conditions tested? Is it possible that 24hrs are not enough?
- However PAX6 protein expression seems to decrease. What is their hypothesis?
- It would have been interesting testing the RAR/RXR level expression in these cells before/after performing the drug-treatment
Reviewer 2 Report
The manuscript entitled “Similarities in DSG1 and KRT3 downregulation through retinoic acid treatment and PAX6 knockdown: Does PAX6 affect RA signaling in limbal epithelial cells?” by Latta et al. describes the results obtained in experiments in which human limbal epithelial cells (LEC) to four compounds related to vitamin A and vitamin A metabolites. Unfortunately, there are many concerns in this paper which make it not suitable for publication. I will list the most important ones below.
Major comments:
- The English used in this paper is so bad that it is hard to understand many phrases. The grammar is often wrong, but also the words are used in a wrong meaning. For example “charge” is used instead of “loading”, “expect” instead of “except”, “invested” instead of “investigated”, etc.
- The title suggest that the research was done using cells in which PAX6 was knocked down, but this was not the case.
- The abstract should not contain the exact values and p levels. Instead it should contain general summary of the results obtained.
- The keywords section should not contain abbreviations.
- The Introduction is very chaotic and it is hard to understand the background for the study.
- Many genes are presented only by their abbreviations. Full names of genes, as well as proteins should be presented at their first appearance.
- All-trans-retinoic acid should be abbreviated by ATRA or at-RA, not RA.
- AGN193109 is not a “blocker” of retinoic acid receptor (RAR). It is a pan-antagonist to all forms of RARs (α, β and γ).
- Similarly, UVI3003 is not a RXR-blocker but an RXR antagonist.
- Antibodies listed in Table 2 are most probably anti-human, not anti-mouse or anti-rabbit, as given.
- In the first paragraph of results section (lines 184-187), the authors said that they estimated toxic effects of AGN193109 and UVI3003 by changes in expression of reference genes. This is an error in the methodology. Toxicity of the compounds should be estimated before starting the proper experiments, by using adequate assays such as cell counting after exposure to the compounds, MTT assay, XTT assay or BrdU incorporation etc. Selection of proper reference gene should be also performed. Expression of reference genes should be stable upon treatment, and there are many algorithms that help to select the proper ones.
- In lines 205-210 the authors write that they present data from 2 western blot measurements, which gave diverse results. Such results should be rather repeated, not presented.
- The figure captions should not contain description of the results obtained, which was repeated in the results.
- Section 4.2 caption suggest that cell cycle has been studied, what was not the case.
- The Discussion section contains many unjustified conclusions.
- Lines 393-394 contain a sentence in German.
Reviewer 3 Report
The manuscript attempts to address new issues regarding the potential link between the protein PAX6 and the retinoic acid signaling in limbal epithelial cells.
The title and abstract are informative and give a clear idea of what to expect from the paper. The main text is a bit difficult to read.
Regarding the experimental part, this study mainly employed cellular assays consisting in the quantification of mRNA expression of several targets under various treatments. This study may be interesting and does bear novelties but the subject matter is very specific and aimed at a limited audience. The authors do not over-interpret the results and have the wisdom to remain cautious in their conclusions.
However the following is a list of few points that should be clarified and/or improved before publication to make the conclusions more sound:
The numerous variations observed for the different genes studied are difficult to interpret and make reading the manuscript tedious. This point is reinforced by the type of ligands used for the treatments and the ligand concentrations chosen. For instance, all-trans retinoic acid (ATRA) is used at 1µM and 5µM. At such concentrations ATRA can activate pathways other than those regulated by the receptor RAR. On the other hand, a commercial solution of RA contains all-trans retinoic acid, but also some 9cisRA (an isomer formed by light) which has the property of activating RXR, the heterodimerization partner of RAR, and thus over-activating the RXR-RAR heterodimer. Although the question remains controversial, such isomerisation of ATRA to 9CRA has not been demonstrated in the body or cells. Therefore the use of a 5µM ATRA solution does not completely reflect the accumulation of ATRA in the cell. Also the use of ATRA at 0.1µM or a synthetic RAR agonist (e.g. TTNPB) would make it possible to discriminate between regulations which are specific or not of RAR.
In the same spirit, it is a question of being careful with the antagonists (term which is preferable to 'blocker' used by authors) RAR (AGN193109) and RXR (UVI3003). AGN is indeed a RAR antagonist, but is defined as an inverse agonist, i.e. it induces a reinforcement of the interaction of transcriptional co-repressors with RAR, and therefore an increase of the transcriptional repression, which may explain the effects exerted by this molecule in the experiments presented in this study. Concerning UVI, this compound is an effective RXR antagonist, but at concentrations higher than 1µM, UVI can activate the PPARG nuclear receptor (NR1C3) (Zhu et al. Toxicology and applied pharmacology, 2017, Vol.314, p.91-97). It would have been interesting to use UVI at 0.1 or 0.3 µM. At least, it would have been desirable for the authors to comment on this.
Authors use trivial names for RAR, as well for other nuclear receptors. A logical numbering system and receptor code, supporting the trivial names, was made by the International Committee of Pharmacology Committee on Receptor Nomenclature and Classification (NC-IUPHAR). In each manuscript dealing with nuclear receptors, it is recommended that the receptors be identified by the official names at least once in the summary and the introduction. Once the name has been established, authors may use the trivial name for the remainder of the manuscript. For instance, the trivial names and the formal nomenclature for RAR is NR1B, RXR is NR2B, and PPAR is NR1C.
Reviewer 4 Report
The manuscript „Similarities in DSG1 and KRT3 downregulation through retinoic acid treatment and PAX6 knockdown: Does PAX6 affect RA signalling in limbal epithelial cells? “ written by Latta L, Knebel I, Bleil C, Stachon T, Katiyar P, Zussy C, Fries FN, Kasmann-Keller B, Seitz B and Szentamary N. describes the effects of retinol, retinoic acid and inhibitors of retinoic acid signalling on limbal epithelial cells. The aim of the investigation was to compare the effects of retinoic acid and modulations of this pathway with the consequences of PAX6 dysregulation, leading to aniridia development.
The manuscript presents interesting data and well organized experiments on primary cell cultures, but should be reorganized and shortened.
In the Abstract, all abbreviation should be explained. Abstract could be shortened and details on the drug concentrations and fold changes in gene expression could be omitted.
Introduction is also too long and too detailed. Pax6 is considered a master transcription factor and has many downstream signalling pathways. It could be mentioned that it can be upregulated by retinoic acid and neurogenin, but all these regulation can be also independent of each other and, on the other side, be additionally regulated by feed back loops and a whole network of other signalling pathways. Also, retinoic acid can act as a morphogen and influence global expression network which can be different in different types of cells, and therefore, downstream target genes can vary. Therefore, i. e. differences in keratin expression in the skin are logical and there is no use of mentioning them. The last three paragraphs of Introduction can be shortened.
Full names of genes should be given: i. e. KRT is keratin.
Material and methods
Table 1 can be put in the Appendix and the data from Table 2. can be written in the text.
In Results, a short introduction for each experiment could be written, describing the aim and basic experimental settings and explaining what and why was done. Also, full names of some genes should be mentioned (i. e. DSG: desmoglein). It could be explained why i. e. SPINK7 was tested.
Cells were treated during 2 days and possibly for differentiation changes at least 5 day treatment could be necessary.
In figures, only figure description should be given and explanations and discussion should be put in results or discussion (i. e. second paragraph in Figure 1. and in other figures. It is not necessary to write that some samples were downregulated or similar.)
Discussion should be shortened, united in one section, and could have a short introduction.
PAX6 and retinoic acid can act directly, but also there are many secondary responses. They belong to the whole network of specific interactions, often involved in feed back loops. In the manuscript some individual signalling pathways are too much accented (such as AP1), as well as changes in downstream genes, and other (such as global networks and interactions which make specific intracellular milieu) are not mentioned. Keratins and desmoglein i. e. are probably downstream targets, reflecting different differentiation pathways initiated. I would suggest connecting data with more global mechanisms of differentiation, to put them in the context of regulatory network directing development, considering other signalling pathways interacting with PAX6 and RA (i. e. articles from Cvekl A) and omitting details.
Table 3 is very useful, but all LEC (si PAX data) can be written together in one column (possibly with a mark explaining the origin of data).
Discussing siADH it is mentioned that there was no changes in protein level. In that case it is not a functional knockdown.
5 µM is used for in vitro differentiation and it is not considered toxic.
Conclusion is also too long, with details which are not necessary.
Minor comments:
Ref. 17 is not complete
Units should written separated from the number.
Line 391: correction comment left
Reviewer 5 Report
Dear authors,
in the manuscript by Lorenz Latta et al., you describe the in vitro systems to demonstrate that in cells with a reduced PAX6 dosage, an altered RA metabolism could be one of the pathways driving transcriptional changes, resulting in a defective corneal epithelial differentiation phenotype. In this manuscript, you reveal similarities in RA induced transcriptional changes and the transcriptional profile of aniridia ocular epithelial cells in order to build a hypothesis of AAK pathogenesis, related to RA metabolism.
I consider this work interesting, and I think it deserves another effort to get better. My comments are as follows:
- The authors should report the abbreviations only the first time, and then use the abbreviation only (for example, see lines 62-64).
- The authors should check the correct name of the abbreviation. For example, PPARs are Peroxisome proliferator-activated receptors and not peroxisome proliferating factors as you report in line 70.
- The authors reported their studies carried out at different concentrations. Sometimes you used 0.5, 1.0 and 1.5 micromolar and sometimes 1 and 5 micromolar. Why did you not use the same concentration for all tests?
- In the Results paragraph, you report “As AGN and UVI treatment for 48h led to expression changes (probably due to toxic side effects) of the GUSB and TBP reference genes (used for normalization of each target gene), we do not display the 48h AGN and UVI treatment results at the figures and these were also not included in our statistical analysis, in order to avoid misinterpretations” (line 184-187). Can you better explain this concept? Why, in your opinion, there are these toxic side effects and why these induce the expression changes?
- In the Conclusions, you report “the hypothesis generated within this study should be validated at metabolic and proteomic level, in aniridia patient cells, in the future”. Could the authors include in this manuscript other studies to validate your hypothesis?